# Dreamitate: Real-World Visuomotor Policy Learning via Video Generation

**Junbang Liang**[*1]    **Ruoshi Liu**[*1]    **Ege Ozguroglu**[1]    **Sruthi Sudhakar**[1]
**Achal Dave**[2]    **Pavel Tokmakov**[2]    **Shuran Song**[3]    **Carl Vondrick**[1]

[1]Columbia University    [2]Toyota Research Institute    [3]Stanford University

dreamitate.cs.columbia.edu

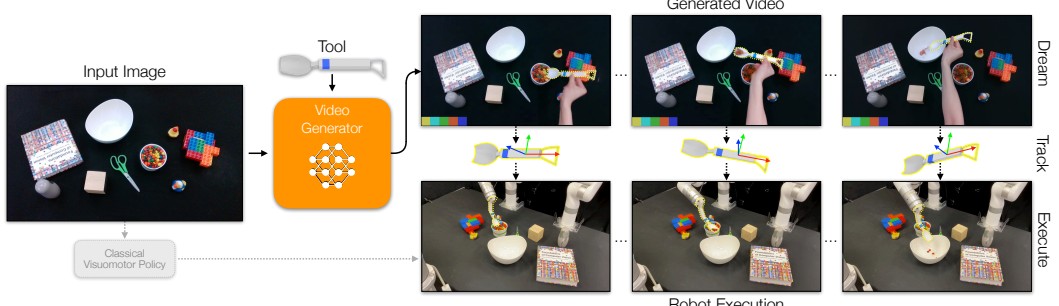

Figure 1: **Real-World Visuomotor Policy Learning via Video Generation.** Dreamitate is a visuomotor policy learning framework that fine-tunes a video generative model to synthesize videos (indicated by ▭) of humans using tools to complete a task. The tool's trajectory in the generated video is tracked, and the robot executes this trajectory to accomplish the task in the real-world.

**Abstract:** A key challenge in manipulation is learning a policy that can robustly generalize to diverse visual environments. A promising approach for learning robust policies is to leverage video generative models, which are pretrained on large-scale datasets of internet videos. In this paper, we propose a visuomotor policy learning framework that fine-tunes a video diffusion model on human demonstrations of a given task. At test time, we generate an example of an execution of the task conditioned on images of a novel scene, and use this synthesized execution directly to control the robot. Our key insight is that using common tools allows us to effortlessly bridge the embodiment gap between the human hand and the robot manipulator. We evaluate our approach on four tasks of increasing complexity and demonstrate that harnessing internet-scale generative models allows the learned policy to achieve a significantly higher degree of generalization than existing behavior cloning approaches.

**Keywords:** Visuomotor Policy Learning, Imitation Learning, Video Generation

## 1 Introduction

Learning visuomotor policies that decide actions from perception has been a longstanding challenge in robotics because they need to generalize to diverse physical situations [1]. Recently, behavior cloning from human demonstrations [2, 3] has become the method of choice for many manipulation tasks, where visuomotor policy learning is formulated as a regression problem to map visual observations to actions supervised by human demonstrations. However, behavior cloning requires ground truth robot actions, which makes scaling to diverse situations challenging.

---

* Equal contribution.

8th Conference on Robot Learning (CoRL 2024), Munich, Germany.

We aim to learn generalizable visuomotor policies by leveraging video generation. Trained on large-scale datasets of internet videos, video generation provides a promising avenue for generalization because extensive priors from human behavior can be transferred into robot behavior. Many recent methods have also explored how to capitalize on these priors, such as synthesizing videos of human behavior [4] or synthesizing videos of robot behavior [5]. However, while videos of people capture the diversity of behavior, generated human actions are difficult to transfer to robots due to the embodiment gap. While directly synthesizing videos of robots would be more realizable physically, the scale of available data is many orders of magnitude smaller than videos of humans, making the learned policy less robust in an in-the-wild environment.

How else can we leverage video generation for policy learning? In this work, we introduce Dreamitate,[1] shown in Figure 1, a visuomotor policy that controls robot behavior via conditional video generation. Our key insight is that the end-effector is the most important part of the embodiment for manipulation: its interaction with diverse objects is challenging to model explicitly and can benefit most from common sense knowledge from video generation. The rest of robot embodiment can be solved through inverse kinematics. Given a visual observation, our policy synthesizes a video of a person performing tasks with a tool. To translate into robot behaviors, we 3D track the tool in the synthesized video, and simply transfer the trajectory into explicit robot actions.

Our formulation offers several major advantages compared to traditional visuomotor policies:

- **Generalizability.** Our underlying video generation model is pretrained on a massive amount of videos available on the internet, including a wide variety of human manipulation in all kinds of environments. By formulating policy learning as video model finetuning, the prior knowledge learned from pre-taining can be preserved, allowing our model to learn generalizable manipulation skills with the same amount of demonstration data.

- **Scalability.** Our finetuning videos are recordings of human demonstrations instead of teleoperation, making our dataset collection more scalable.

- **Interpretability.** During inference time, our video model predicts future execution plans in the form of videos before the actual robot executions are performed. Compared to a black-box end-to-end policy, our formulation offers an intermediate representation of policy that is interpretable to humans, a key feature for human-robot interaction applications.

We evaluated our approach across four real-world tasks, including bimanual manipulation, precise 3D manipulation, and long-horizon tasks, using a small number of expert human demonstrations. We found that the video model consistently outperformed the baseline behavior cloning model in generalizing to unseen scenarios. Additionally, we analyzed the model's performance when scaling down the training dataset and found it maintained strong generalization performance even with fewer demonstrations. We will release the code and data for reproducing our results.

## 2  Related Work

**Behavior Cloning.** Behavior cloning (BC) has emerged as a leading approach in manipulation tasks, learning policies from demonstrations through supervised learning. Early BC efforts involved end-to-end models mapping states to actions, but struggled with multimodal behaviors and high-precision tasks [6, 7, 8]. Subsequently, Energy-Based Models (EBMs) were explored, predicting actions by minimizing energy in sequence optimization [9, 10, 11, 12]. Recently, conditional generative models have shown promise in capturing multimodal demonstration behaviors, enhancing task success rates [2, 3, 13]. Different from previous models, we integrate video prediction and 3D tracking to predict action prediction.

**Visual Pretraining for Policy Learning.** Prior works have extensively explored various ways to pre-train the perception model in a visuomotor policy for learning more robust visual representation.

---

[1]dream·i·tate (verb) \drēm e tāt\. To behave in a way that imitates a dream.

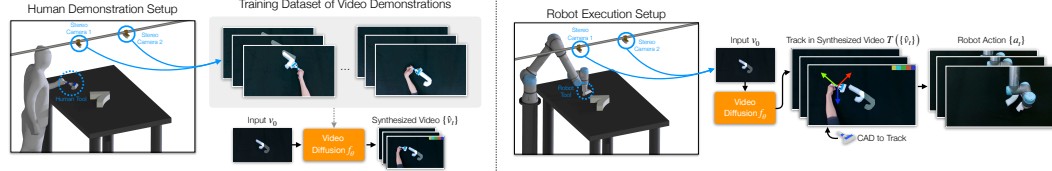

Figure 2: **Method Overview.** For each task, (Step 1) we capture stereo camera recordings of human demonstrations of tool use from a top and a side view. (Step 2) The video model is fine-tuned to recreate these demonstrations from an initial scene image. (Step 3) In a new, unseen scenario, a stereo image pair is provided to the video model to generate (▭) the manipulation. The tool's trajectory in the generated video is tracked and executed by the robot to complete the task.

One of the most popular pertaining objectives has been video prediction [14, 15, 16, 17, 18]. By learning to predict future based on the current observation, a model can learn the dynamic and causality of the world crucial for physical interaction. Contrastive learning [15, 19, 20, 21] as well as masked autoencoding [22, 23, 24] are two very popular self-supervised learning objectives for learning visual representation for robotics. Besides, another line of work [25, 26, 27] studies learning a generalizable reward function through visual pertaining for reinforcement learning.

**Video Models for Decision-Making.** More recently, in light of the fast progress of text-to-video generative models [28, 29, 30, 31, 32], the hope to leverage internet-scale video pertaining for robotics is rekindled [33]. A line of work uses video generative model as a world simulator [34, 5, 28], which can predict future video conditioned on an action. Another line of work uses video-language model for longer-horizon planning [35, 36, 37]. Different from prior works on this topic which typically use video prediction models as a world model or simulator in order to perform planning and decision making, Dreamitate uses video prediction model as part of a visuomotor policy and directly uses the predicted video for action prediction.

## 3 Approach

### 3.1 Overview

Given a visual observation of the scene $v_0$, our goal is to plan and execute robot actions $a_t \in \text{SE}(3)$ via video generation. We approach this problem through the framework, visualized in Figure 2:

$$a_t = \mathcal{T}(\hat{v}_t) \quad \text{where} \quad \{\hat{v}_t\} = f_\theta(v_0)$$

where $f_\theta(\cdot)$ is a generative video model with learnable parameters $\theta$. We train $f$ to generate videos of a human doing the task using a tool that we can track in 3D. $\mathcal{T}$ tracks the trajectory of the tool in the generated video, which we can directly use to control the end-effector state to perform manipulation tasks. This com-

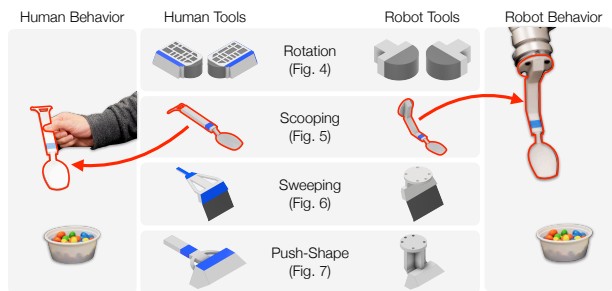

Figure 3: **Aligning Human and Robot Behavior via Trackable Tools.** We use tools with known CAD models for human demonstrations to facilitate precise 3D tracking across four real-world manipulation tasks. We design unique tools (e.g., a spoon for scooping) to show the generalizability of our approach for varying tasks.

position of synthesize-then-track facilitates cheap data collection without teleoperation, and aligns with pretaining video data, which is composed of mostly videos showing human behavior.

### 3.2 Video Generation

Starting with a pre-trained video generator, we fine-tune it on a small video demonstration dataset in order to combine the priors from internet videos with behaviors from demonstration videos. To

collect videos for fine-tuning, we build a tabletop setup with two calibrated cameras positioned 45 degrees apart for better visibility. We capture several pair of stereo videos $(\hat{v}^1, \hat{v}^2) \in \mathcal{V}$ of humans performing the task using a 3D printed tool (see Figure 3 for details).

We use these demonstration videos to fine-tune the video generative model. Since we will imitate the trajectory of the synthesized tool in the physical world, we generate stereo videos to make action possible in 3D. Let $(\hat{v}_t^1, \hat{v}_t^2)$ be the predicted stereo video frames at time $t$ and $(v_t^1, v_t^2)$ be the ground truth stereo frames from demonstration videos. We optimize the video prediction objective:

$$\min_\theta \mathbb{E}_{v \in \mathcal{V}} \left[ \sum_{t=1}^{T} ||(\hat{v}_t^1 - v_t^1)||_2 + ||(\hat{v}_t^2 - v_t^2)||_2 \right] \quad \text{for} \quad \{\hat{v}_t\} = f_\theta(v_0) \tag{1}$$

We initialize $\theta$ to the pre-trained weights learned from large-scale internet video datasets (Stable Video Diffusion [29]). We train a separate $f_\theta$ for each task (e.g. sweeping or scooping).

Following the implementation from [38], the encoder and decoder are frozen such that only the spatial/temporal attention layers are fine-tuned. The per-frame image embedding input to the model is modified based on the viewing angle of the output frame to facilitate stereo video generation. This ensures that the first half of the frames generate the first view and the half generates the second view, using the initial image embedding from the respective view.

At test time, we will be given a stereo image pair of the scene that we wish the robot to manipulate. We apply the fine-tuned video model $f(v_0)$ and obtain a generated stereo video frames $\hat{v}_t$ of the task being performed.

### 3.3 Track then Act

The synthesized video frames $\{\hat{v}_t\}$ serve as the intermediate representation of our policy to obtain an action trajectory, $\{a_t\}$, that we then execute on the robot. Actions $a_t \in \mathrm{SE}(3)$ are represented as the 6D pose of the tool relative to the camera. By using a known CAD model, we can efficiently and accurately track the tool in each frame of the generated videos and therefore obtain precise locations of the tool. Each action $a_i$ in the action trajectory corresponds to one frame from the synthesized video. For 3D consistency, we use the generated stereo pair, along with the calibrated camera parameters to resolve 6D pose of the tool. For tracking the tool from an RGB image, we use Megapose [39] and operate on $768 \times 448$ resolution video frames with camera intrinsic derived from the default values of the Intel Realsense camera. Finally, during execution, the customized tool is mounted on the robot arm and actions $a_0, ..., a_T$ are executed by the robot with the trajectory smoothed using impedance control. The entire data-collection, training, and fine-tuning pipeline are agent-independent. This is possible because we use tool's as manipulators and track known tools in generated videos, removing the need for learning any agent-specific action spaces.

## 4 Experiments

In this section, we evaluate the performance of our video model in real-world robot manipulation tasks. The evaluation, summarized in Table 1, encompasses four distinct tasks including object rotation, granular material scooping, table top sweeping and shape pushing (see Figure 3 for a visualization of the used tools). The details of each task are further explained in the corresponding sections below. For an assessment of our model's capabilities, we benchmark its performance against Diffusion Policy [40], Action Chunking Transformer (ACT) [3] and Vector-Quantized Behavior Transformer (VQ-BeT) [13] as baselines.

### 4.1 Experimental Setup

To evaluate the model's generalizability, the data was collected by a human demonstrator at different locations for training and evaluation using the same camera setup, though the exact tabletop and lighting conditions varied. To further test the model's robustness in real-world scenarios, we ensured

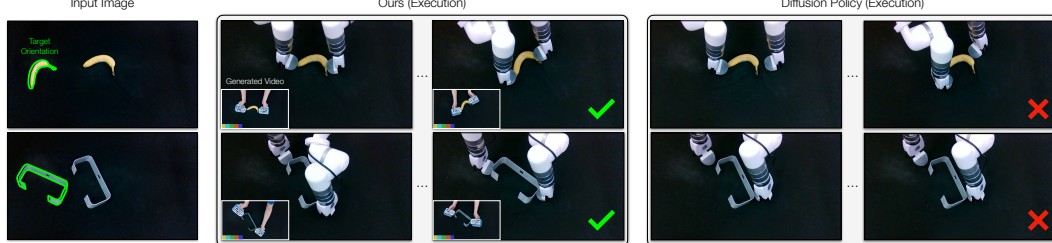

Figure 4: **Rotation Qualitative Results.** For this task, we provide as input the image on the left for each row, with an example of a successful rotation overlaid. Our video generation-based approach succeeds, while Diffusion Policy fails to select stable grasping points on the object, causing it to slip during manipulation. Generated videos are indicated by ▨.

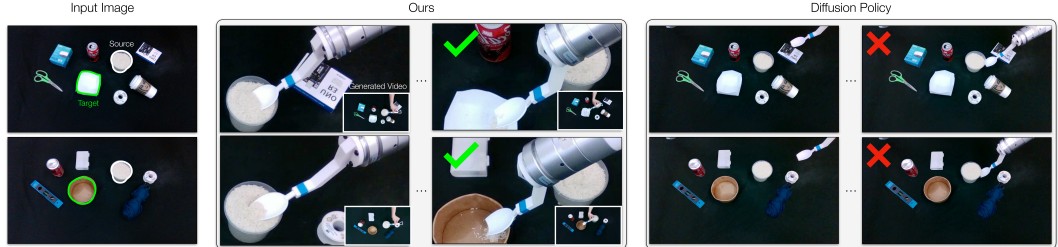

Figure 5: **Scooping Qualitative Results.** We provide the input image on the left (without the overlays provided for illustration) to our model and Diffusion Policy, and show the output trajectory on the right. Our approach succeeds in most trials, while Diffusion Policy is often distracted by other objects in the scene, placing material in the wrong container and failing to scoop accurately. Generated videos are indicated by ▨.

that the sets of objects used in training and evaluation are non-overlapping. To initialize tool tracking with Megapose [39], background subtraction and hand removal based on skin color were employed to obtain the tool bounding box. In rare cases where this process failed, human corrections were applied to the bounding boxes (both for our method and for the baseline).

We used the same training data to train the baselines with stereo image input. The videos were preprocessed using Megapose for tool tracking, providing target trajectories for training. For Diffusion Policy we used a pretrained ResNet-18 variant [40] as the baseline, as the pretrained CLIP encoder [41] variant showed lower performance in our tasks. We trained the baselines to predict the entire action trajectory for the next 12 time steps without replan as an open-loop system, similar to the video model.

| | Training Objects | Demonstrations | Test Objects | Test Trials |
|---|---|---|---|---|
| Rotation | 31 | 371 | 10 | 40 |
| Scooping | 17 Bowls, 8 Particles | 368 | 8 Bowls, 4 Particles | 40 |
| Sweeping | 6 Particles | 356 | 6 Particles | 40 |
| Push-Shape | 26 Letters | 727 | 8 Shapes | 32 |

Table 1: **Tasks Summary.** We detail the train and test setup for each task above. For each task, we use a distinct set of objects during testing than training.

## 4.2 Object Rotation Task

**Task.** We design a rotation task, shown in Figure 4, to test our policy's ability to coordinate end-effectors and choose appropriate grasping points to manipulate real-world objects. In each training demonstration, we randomly place an object on the table, and use a gripper, shown in Figure 3 (top), to grasp the object at appropriate points and rotate it counterclockwise up to 45 degrees.

**Setup.** We collect training data with 31 objects, with 14 real-world objects (such as boxes or a hammer) and 17 custom colored shapes made out of foam. We use 10 unseen real-world objects for

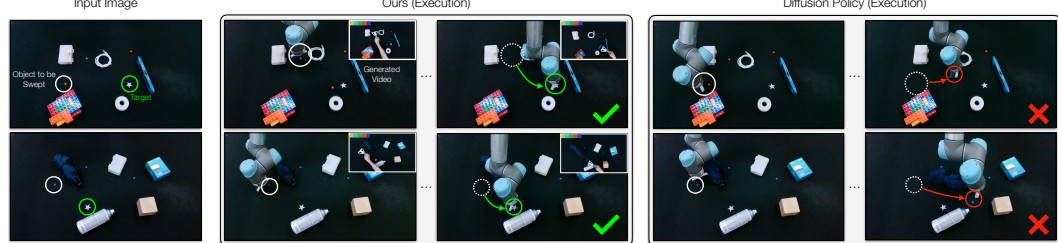

Figure 6: **Sweeping Qualitative Results.** We provide the input image on the left (minus the white and green circles overlaid for illustration) to our model and Diffusion Policy, and show the output trajectories on the right. Our approach again achieves high success rate even on this challenging, multi-modal task, whereas Diffusion Policy generates trajectories that often collide with obstacles and fail to sweep to the target. Generated videos are indicated by ▭▭▭▭.

|  | Rotation | Scooping | Sweeping | Push-Shape | |
| Model | Success Rate | Success Rate | Success Rate | mIoU | Rot. Error |
|---|---|---|---|---|---|
| ACT [3] | 5 / 40 | 10 / 40 | 3 / 40 | 0.527 | 44.1° |
| VQ-BeT [13] | 4 / 40 | 11 / 40 | 0 / 40 | 0.477 | 47.4° |
| Diffusion Policy [2] | 22 / 40 | 22 / 40 | 5 / 40 | 0.550 | 48.2° |
| **Ours** | **37 / 40** | **34 / 40** | **37 / 40** | **0.731** | **8.0°** |

Table 2: **Quantitative Results.** We compare our method to ACT, VQ-BeT and Diffusion Policy on four tasks quantitatively. We report success rates (successful trials / total trials) for rotation, scooping, and sweeping tasks. For Push-Shape, we report mean intersection-over-union (mIoU) and average rotation error (Rot. Error). Our approach performs well across all tasks, whereas baselines shows worse performance overall and degrades in the more challenging sweeping and Push-Shape scenarios.

evaluation. We mark a trajectory as successful if the robot makes and maintains contact with the object throughout the rotation, and rotates it at least 25 degrees counterclockwise.

**Results.** As shown in Table 2, our policy significantly outperforms baselines (92.5% vs. 55% at best). Fig. 4 illustrates cases where our policy succeeds while Diffusion Policy fails. We observe that baselines can fail to move the end-effector into contact with the object. In more challenging cases, baselines are prone to selecting unstable grasping points causing the object to slip during manipulation. In contrast, our policy consistently makes contact with the object, with limited failures when selecting appropriate grasping points for particularly challenging shapes, such as a transparent bag with toys inside.

### 4.3 Granular Material Scooping Task

**Task.** This task, shown in Figure 5, requires scooping granular material (e.g., beans) from a full container to an empty one while avoiding distractor objects. This task requires the policy to perform precise manipulation of a scooping tool, identify the full and the empty containers, along with their precise locations, in arbitrary positions, and ignore distractors.

**Setup.** We collect demonstrations with 17 bowls and 8 colored beans with only 1 distraction object in the scene at a time. At test time, we use an unseen set of 8 bowls, 4 new colored particles, and 5 distractor objects per trial sampled from a fixed set of 15 everyday objects. We mark a trajectory as successful if the robot transfers any particles to the empty bowl.

**Results.** As shown in Table 2 (middle), our policy significantly outperforms baselines (85% vs. 55% at best). This is a particularly challenging task for our approach, due to the small target (the scooper) for object tracking and video generation. This demonstrates that stereo video generation can accurately determine the object's pose to perform 3D manipulation. By comparison, baselines are often distracted by objects in the scene, placing material in the wrong container, and failing more frequently in scooping due to misjudging the bowl's height on the table, as illustrated in Fig. 5.

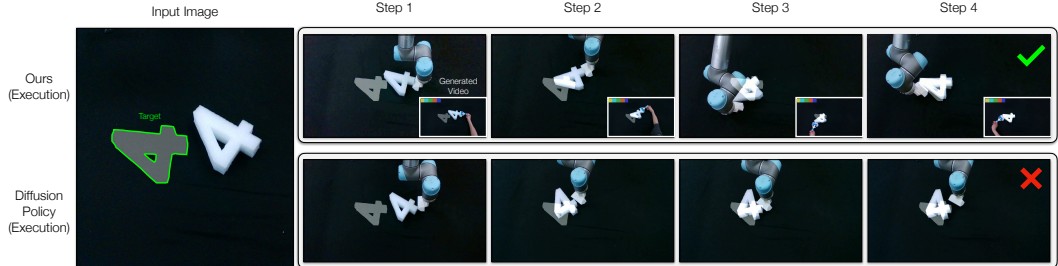

Figure 7: **Push-Shape Qualitative Results.** We provide the image on the left, including the gray overlay indicating the target object position and orientation, to our model and Diffusion Policy, and show output trajectories on the right. Our method produces appropriate pushing actions to adjust the object's position. In contrast, Diffusion Policy fails to effectively rotate the object to match the mask. Generated videos (top row, bottom right) are indicated by ▮▮▮▮▮.

## 4.4 Table Top Sweeping Task

**Task.** The sweeping task, shown in Figure 6, requires policies to use a brush to sweep randomly placed particles to a target location marked by a randomly placed star, while avoiding obstacles. This task is designed to test policy's ability to handle multi-modal distributions in the training data.

**Setup.** Training data includes 6 colored particles and 25 distraction objects, with 5 to 6 distractions at a time. At test time, we use 6 new particles and 15 unseen distractions, with 2 to 4 beans and 5 distractions randomly placed on the table. The training data contains some variations with multiple ways to achieve the goal, such as choosing which particle to sweep in the scene. Due to the small size of the particles, we remove the final pooling layer from Diffusion Policy ResNet-18 encoder to provide a higher spatial resolution and improve baseline performance. We mark a trajectory as successful if any particle is transferred to be within 50 mm from the target.

**Results.** In Table 2 (right) we observe that the video model maintains strong performance in this task with a 92.5% success rate. In contrast, baselines generate trajectories that often collide with obstacles and fail to sweep to the target (see Fig. 6), achieving only a 12.5% success rate at best with Diffusion Policy. This experiment demonstrates that the capitalizing on internet-scale pre-trained video generation models allows to better handle multi-modal demonstrations and achieve a much larger degree of generalization in this challenging scenario.

## 4.5 Push-Shape Task (Long Horizon)

**Task.** Push-Shape, shown in Figure 7, is a challenging version of! the long-horizon Push-T task [2]: we place a foam shape on the table, and task the robot with pushing the object to a specified target goal mask (given as input to the policy) over consecutive steps. This is a challenging task as it requires adjusting the position and orientation of the shape, requiring predicting the shape's movement, which depends on the contact between the object and the end-effector, as well as material properties of the table.

**Setup.** We train with a set of 26 foam objects each in the shape of a letter from the alphabet, and test on 8 unseen foam shapes (including digits and polygons). As this is a challenging, multi-step task, we score the best of 4 rollouts for each trial, and report the mean intersect-over-union (mIoU) with the target mask as well as the average rotation error from the target.

**Results.** As shown in Table 2, over the 32 trials, the video model achieved a significantly higher mIoU of 0.731 compared to 0.550 mIoU (at best with Diffusion Policy), and an average rotation error of 8.0 degrees compared to 44.1 degrees (at best with ACT). We find that baselines tend to push the object to the target, but fails to effectively rotate the object to match the mask. By contrast, the video model produces appropriate pushing actions to adjust the object's position and orientation, thereby reducing the rotation error. Qualitative examples of the model rollouts are illustrated in Fig. 7.

### 4.6 Performance Scaling Curve

We studied the impact of training set size on the generalization ability of our policy and compared it with Diffusion Policy using the rotation task. Both models performed well with the full dataset. When retrained with two-thirds and one-third of the dataset and tested over 40 trials, the results depicted in Figure 8 indicate that while Diffusion Policy's performance declines significantly with reduced data, our model remains stable, maintaining high success rates even with only one-third of the data. This highlights our model's superior generalization, which can be attributed to the extensive pre-training on Internet-scale video generation models.

### 4.7 Ablation Study

We investigated the impact of pretraining and stereo camera setups on model generalization using the rotation task. The results are shown in Table 3. Without pretraining, the model produces blurry videos with significantly distorted tools, but the tracking model still tracks the tool's pose with stereo projection correcting depth estimation. We also explored the effects of the stereo camera setup. In the stereo input monocular output (SIMO) model, we track only one view of the stereo output in the full model. Additionally, we trained a monocular input and monocular output (MIMO) model. The SIMO model showed a performance decline due to greater depth estimation errors without stereo projection. The MIMO model exhibited an even further reduction in performance, indicating the importance of stereo input conditioning.

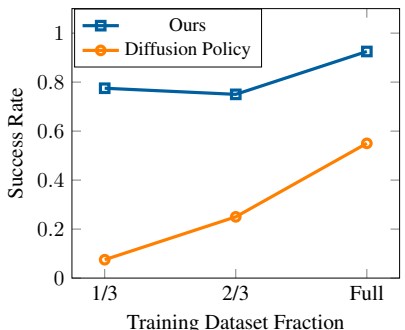

Figure 8: **Number of Demonstrations.** Our approach maintains strong generalization performance with less training data compared to Diffusion Policy.

|  | Rotation |
|---|---|
| Model | Success Rate |
| **Full Model** | **37 / 40** |
| Full Model w/o Pretraining | 18 / 40 |
| Stereo In Monocular Out | 30 / 40 |
| Monocular In Monocular Out | 14 / 40 |

Table 3: **Ablation Results.** We provide results on the effect of pretraining and stereo camera setup on model performance.

## 5 Limitations

Our approach is limited by the need for visually trackable tools, and the model is task-specific, lacking conditional text embeddings to resolve ambiguities in input images. Future work could explore integrating text embeddings to train a unified policy across tasks. The reliance on rigid tools and video generation restricts the model's use in tasks requiring fine-grained force control. Additionally, using tools with known CAD models and camera calibration reduces flexibility, while tracking small objects remains a challenge for pose estimation models, affecting robot trajectory planning. Finally, the computational demands of video models make real-time control impractical, with current processing times of 33.5 seconds per video and 7.5 seconds for object pose tracking on an A100 GPU.

## 6 Conclusion

In this study, we explore how video generative models can learn generalizable visuomotor policies. We propose to fine-tune a video diffusion model on human demonstrations to synthesize execution videos, which are then used to control the robot. Our key insight is that using common tools helps bridge the gap between human demonstrations and the robot manipulator. Experiments show that leveraging pretrained video diffusion models on Internet-scale data leads to greater generalization compared to state-of-the-art behavior cloning methods.

**Acknowledgments**

We would like to thank Basile Van Hoorick and Kyle Sargent for providing the video model training code and for their valuable feedback. We also appreciate the insightful discussions with Paarth Shah and Dian Chen, and the reviewers for their helpful comments. This research was partially supported by the Toyota Research Institute and the NSF NRI Award #2132519.

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

# A    Appendix

## A.1    Video Model Implementation

We adapted the pretrained Stable Video Diffusion model [29], which generates 25-frame videos at a time. In our adaptation, the first 13 frames correspond to the stereo view 1, and the last 12 frames correspond to the stereo view 2, captured from the two cameras. To condition the video model to generate a stereo video, we modified the per-frame image embedding based on the viewing angle of each output frame. Since each frame of the stereo videos should be paired but the video model generates an odd number of frames, the first frame of the video model output is always the same as the input and discarded at test time. The model training hyperparameters are given in Table 4. We conducted the finetuning experiments using four A100 GPUs over a span of 40 hours. During inference, we use 30 denoising steps with a constant classifier-free guidance of 1.0. Additional qualitative results for the four tasks are shown in Fig 9, 10, 11 and 12.

| Hyperparameters | Resolution | Lr | Batch | Steps | Clip Duration | Fps | Motion Score |
|---|---|---|---|---|---|---|---|
| Rotation (Full DS) | 768×448 | 1e-5 | 4 | 16384 | 2.0 | 6 | 200 |
| Rotation (2/3 DS) | 768×448 | 1e-5 | 4 | 16384 | 2.0 | 6 | 200 |
| Rotation (1/3 DS) | 768×448 | 1e-5 | 3 | 15360 | 2.0 | 6 | 200 |
| Rotation (w/o Pretrain) | 768×448 | 1e-5 | 4 | 16384 | 2.0 | 6 | 200 |
| Rotation (SIMO) | 768×448 | 1e-5 | 4 | 16384 | 2.0 | 6 | 200 |
| Rotation (MIMO) | 768×448 | 1e-5 | 4 | 16384 | 2.0 | 6 | 200 |
| Scooping | 768×448 | 1e-5 | 4 | 16384 | 3.0 | 5 | 200 |
| Sweeping | 768×448 | 1e-5 | 4 | 16384 | 3.0 | 5 | 200 |
| Push-Shape | 768×448 | 1e-5 | 4 | 17408 | 2.0 | 6 | 200 |

Table 4: **Hyperparameters for Video Model Training.** Resolution: image and video resolution, Lr: learning rate, Batch: batch size, Steps: training steps for the evaluation checkpoint, Clip Duration: single demonstration video length in seconds, Fps: the video sub-sampling frame rate and model fps parameter, Motion Score: model motion score parameter.

## A.2    Experimental Setup

The stereo camera setup consists of two Intel RealSense D435i cameras spaced approximately 660 mm apart at a 45° angle. The distance between the cameras and the table is about 760 mm. The real-world data collection and the robot experiment setups are shown in Fig. 17 and 18. The training videos are recorded at a resolution of 1280×720 and are then cropped and resized to the appropriate resolution for model input. The table surface used for data collection and experiments is covered with a black cloth, which introduces variations in friction and increases uncertainty in the Push-Shape experiments. For the rotation and scooping experiments, UFACTORY xArm 7 robots are used, while UR5 robots are used for the sweeping and Push-Shape experiments. For calculating the mIoU in the Push-Shape experiment, the view from the stereo camera 1 is used. In each trial with multiple steps, the resulting image with the highest IoU with the target is used to calculate the rotation error. In sweeping and push-shape experiments, the robot end-effector height is limited to avoid robot collision with the table top.

## A.3    Data Collection

For all tasks, the first frame of the human demonstration video is an image of the scene. The subsequent frames include the human demonstrator using the tool to perform the manipulation. In the Push-Shape demonstration, an object is pushed to a location in multiple steps. The final position of the object is used as a mask and blended with the entire video for the target position. The objects used in training and testing for different tasks are shown in Fig. 13, 14, 15 and 16.

### A.4 Object Tracking

In the videos, the tool is tracked using MegaPose [39]. Utilizing a stereo setup, the center of the tracked object from each camera are projected into 3D space as a straight line. The translation component of the object in 3D space is determined by finding the midpoint between the projected lines from the two cameras. The rotation component of the object is obtained by averaging the object rotations from the two views. This refined object pose from the stereo setup enhances the accuracy of the object's depth measurement from the cameras. In the scooping task, only the handle of the scooper is tracked to avoid inaccuracies due to occlusion by particles. In the sweeping and Push-Shape tasks, the tool without the handle is tracked, as the handle is occluded by the human hand. To obtain the tool trajectories for training Diffusion Policy, the same stereo tracking is applied to the demonstration videos.

### A.5 Diffusion Policy Baseline

We use a CNN-based Diffusion Policy as our baseline, employing two pretrained ResNet-18 [42] image encoders to process the stereo images of the scene. The input images have a resolution of 384×224, similar to the original implementation resolution. We found that higher resolution input images did not improve model performance. We trained Diffusion Policy for 200 epochs to predict actions for the next 12 time steps.

### A.6 ACT Baseline

We use the ACT implementation from LeRobot [43] as our baseline, utilizing a single pretrained ResNet-18 [42] image encoder to process stereo images of the scene, as the implementation supports multi-image conditioning by default. The input images have a resolution of 768×448. We trained ACT for approximately 90k steps to predict actions for the subsequent 12 time steps.

### A.7 VQ-BeT Baseline

We use the VQ-BeT implementation from LeRobot [43] as our baseline, modifying it to incorporate two pretrained ResNet-18 [42] image encoders to process stereo images of the scene, since the original implementation does not support multi-image conditioning. The input images have a resolution of 768×448. Through experimentation, we found that training the VQ-BeT VAE for approximately 5k steps, followed by training the entire model for around 85k steps, yielded the best results in predicting actions for the subsequent 12 time steps.

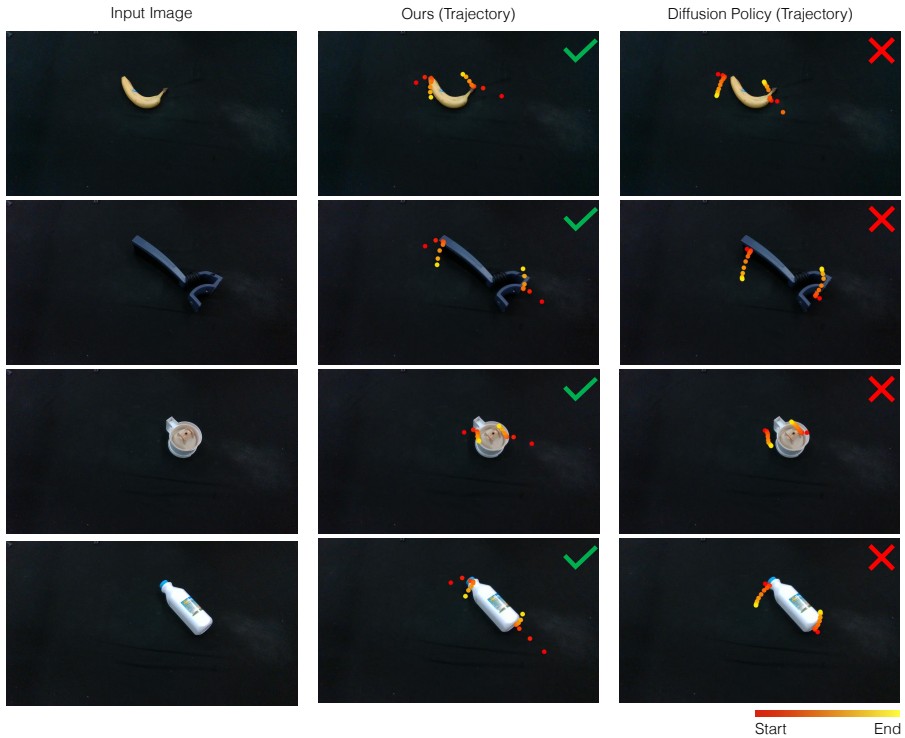

Figure 9: **Additional Rotation Qualitative Results.** The trajectories of the end-effectors are projected onto the input image.

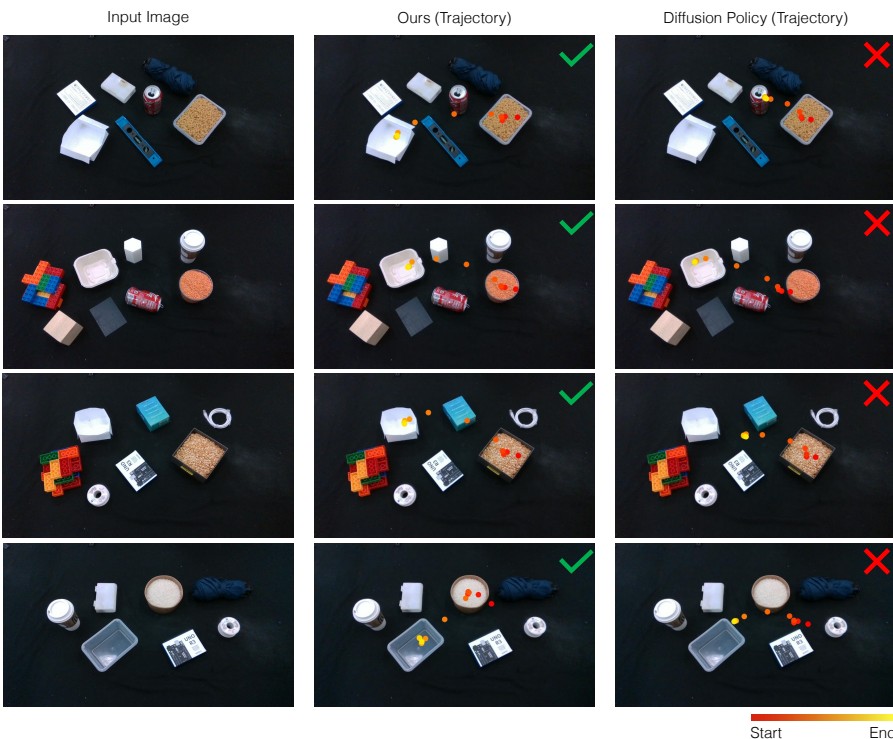

Figure 10: **Additional Scooping Qualitative Results.** The trajectory of the end-effector is projected onto the input image.

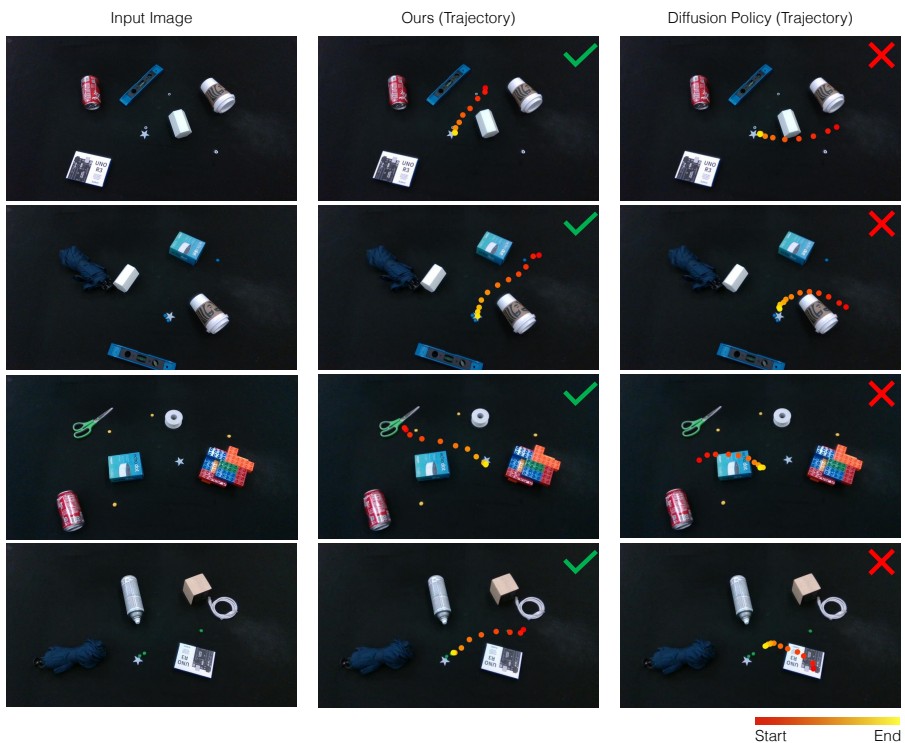

Figure 11: **Additional Sweeping Qualitative Results.** The trajectory of the end-effector is projected onto the input image.

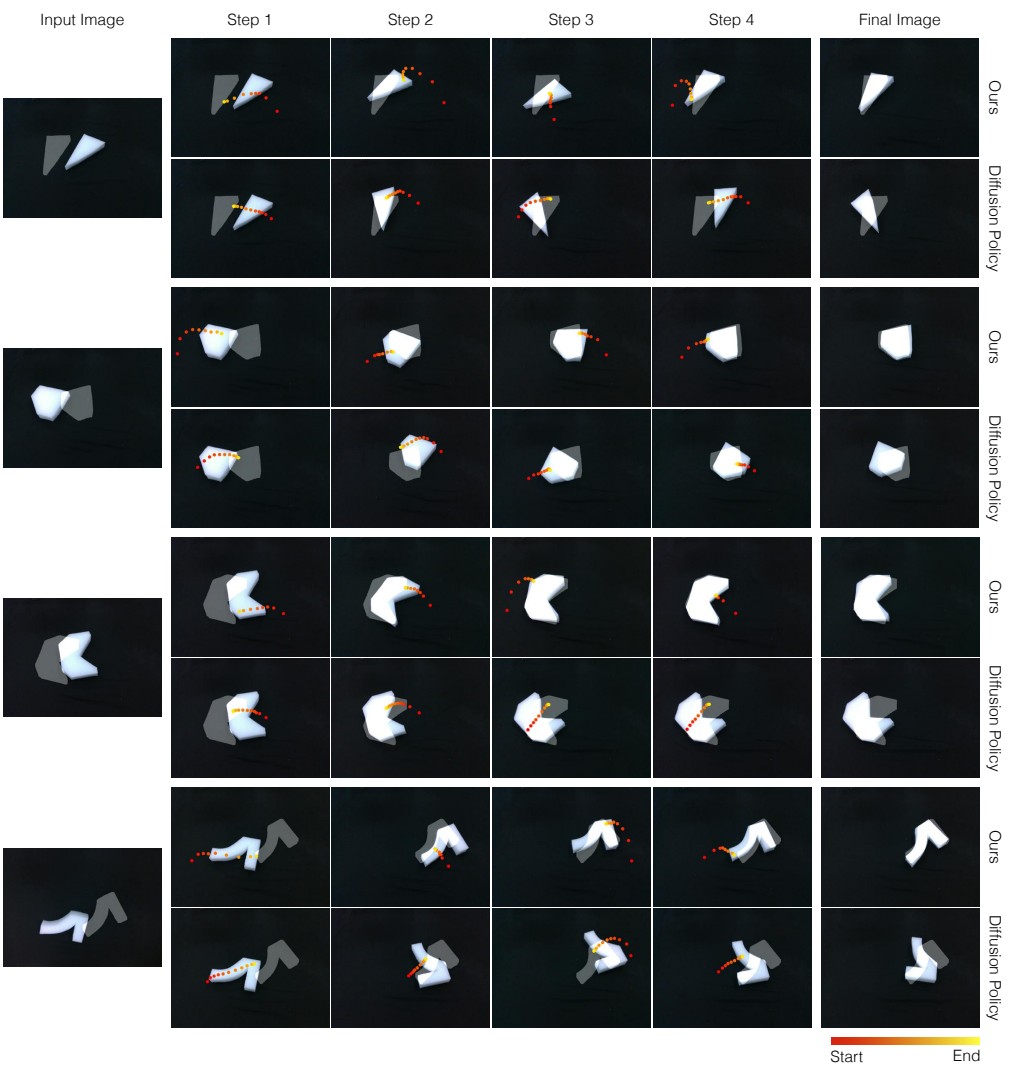

Figure 12: **Additional Push-Shape Qualitative Results.** The trajectory of the end-effector is projected onto the input image.

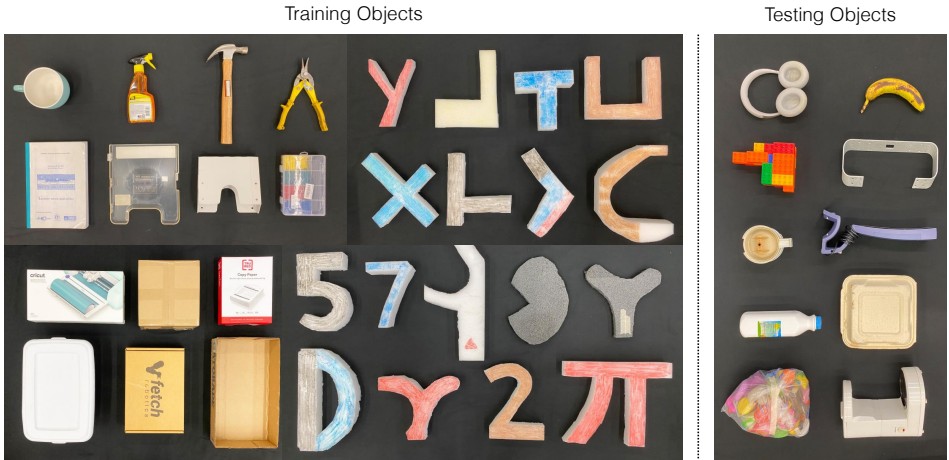

Figure 13: **Rotation Objects.** The training set includes 14 real-world objects and 17 custom colored shapes made out of foam. The testing set includes 10 challenging real-world objects.

Training Objects                                   Testing Objects

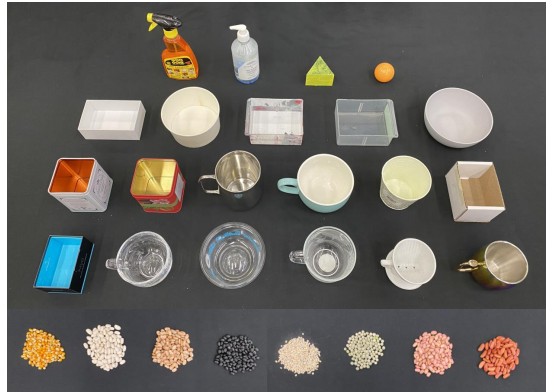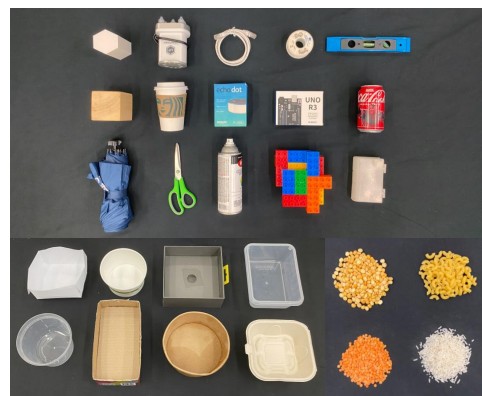

Figure 14: **Scooping Objects.** The training set includes 17 bowls, 8 colored beans and 4 real-world objects. The testing set includes 8 bowls, 4 colored particles and 15 real-world distraction objects. During data collection, both the real-world objects and inverted bowls are used as the single distraction object in the scene.

Training Objects                                   Testing Objects

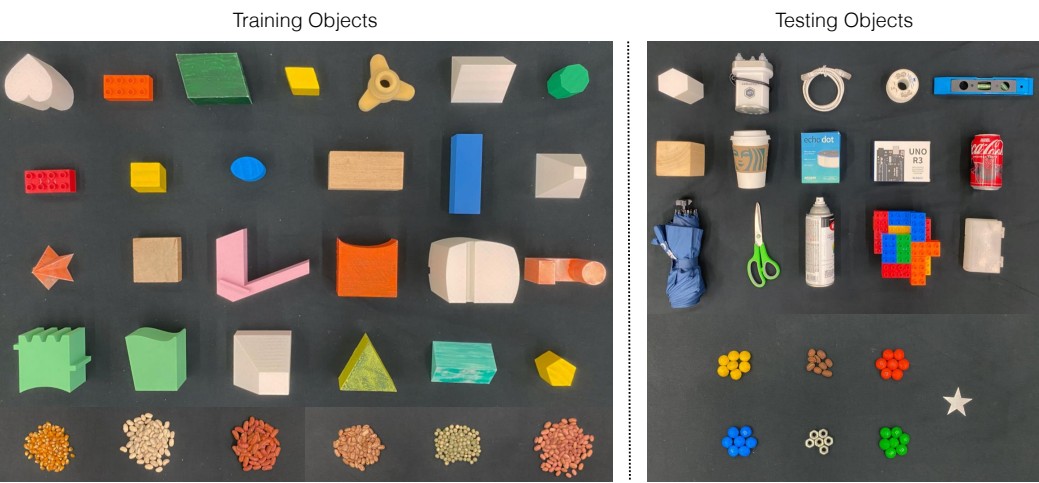

Figure 15: **Sweeping Objects.** The training set includes 25 distraction objects and 6 colored beans, with 5 to 6 distractions in the scene at a time during data collection. The testing set includes 15 real-world distraction objects and 6 colored particles. The star is used as the target in the experiments.

Training Objects                                   Testing Objects

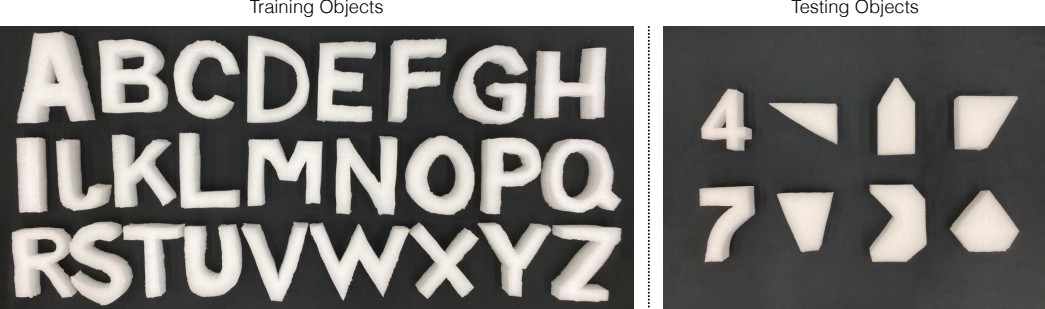

Figure 16: **Push-Shape Objects.** The training set includes all 26 capital letters of the alphabet, while the testing set consists of 8 shapes, including digits and polygons.

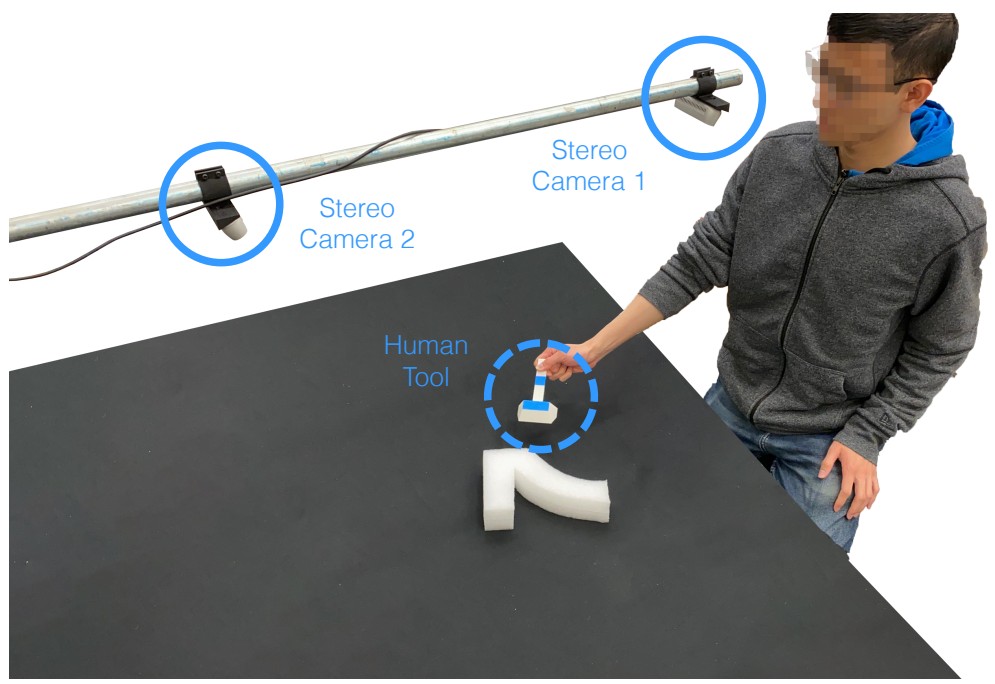

Figure 17: **Real-world Data Collection Setup.** The data collection setup has the same camera arrangement as the robot experiment setup.

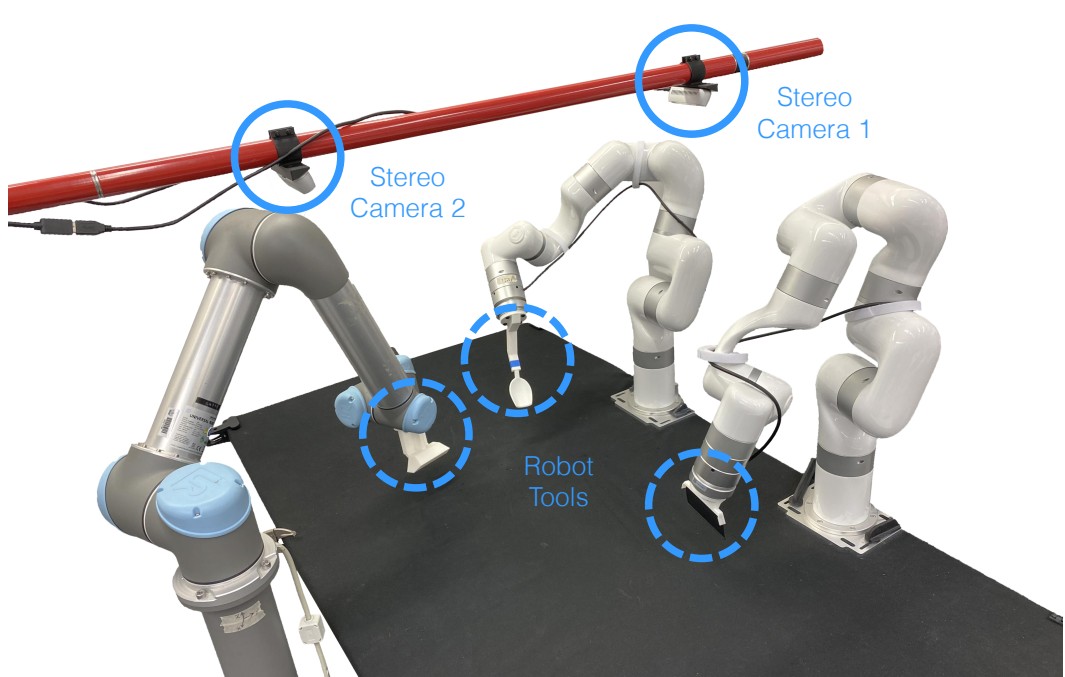

Figure 18: **Real-world Robot Experiment Setup.** The robot experiment setup includes the 3 robots to perform all the experiments.

