# OpenReview forum: "Dreamitate: Real-World Visuomotor Policy Learning via Video Generation"
_robot-learning.org/CoRL/2024/Conference — CoRL 2024_

### Official Review · Reviewer_MoCz · 2024-07-20
**Dreamitate: Real-World Visuomotor Policy Learning via Video Generation**

**Originality:** 4
**Technical Quality:** 3
**Clarity Of Presentation:** 4
**Potential Impact:** 3
**Recommendation:** 2
**Confidence:** 5

**Review:**

### Strengths:
1. **Generalizability:** The use of video generative models pretrained on internet-scale datasets significantly enhances the generalizability of the learned visuomotor policies.
2. **Scalability:** The framework uses human demonstrations for training, which is more scalable than teleoperation-based data collection.
3. **Interpretability:** The intermediate representation of policy as generated videos offers better interpretability compared to end-to-end black-box policies.

### Weaknesses:
1. **Non-Prehensile Tasks:** The evaluated tasks are non-prehensile, limiting the applicability to more complex tasks requiring grasping or picking.
2. **Robustness Testing:** The paper does not thoroughly address how the method handles perturbations during task execution.
3. **Error Propagation:** There is a potential for error propagation in the video generation and tool tracking phase, which can significantly impact the performance of the generated policy.
4. **Lack of Comparative Analysis:** The paper lacks a comprehensive comparison with other state-of-the-art methods beyond behavior cloning, including other hybrid approaches or categorical manipulations such as [1-4].

[1] Peter R. Florence, Lucas Manuelli, and Russ Tedrake. "Dense Object Nets: Learning Dense Visual Object Descriptors By and For Robotic Manipulation."

[2] Leonard Bruns and Patric Jensfel. "SDFEst: Categorical Pose and Shape Estimation of Objects from RGB-D using Signed Distance Fields."

[3] Bowen Wen, Wenzhao Lian, Kostas Bekris, and Stefan Schaal. "You Only Demonstrate Once: Category-Level Manipulation from Single Visual Demonstration."

[4] Lucas Manuelli, Wei Gao, Peter Florence, and Russ Tedrake. "kPAM: KeyPoint Affordances for Category-Level Robotic Manipulation."

**Quality Of The Limitations Section:**

2

**Questions For Rebuttal:**

1. Task Complexity: All the tasks are non-prehensile. How does the method perform on tasks that require picking or grasping?
2. Failure Cases: The tasks can be solved by the proposed method, but what are the typical failure cases for your method and the diffusion policy?
3. Performance Comparison: The Push-Shape task appears similar to the push-T task in the diffusion policy paper. The performance in the Push-Shape task is much worse. Why?
4. Training Data Subsampling: What does "open-loop system without subsampling the training data" in line 155 mean?
5. Task Difficulty: The hardest task, Push-Shape, has been well-solved with both learning-based and model-based methods. Can you perform more realistic and harder tasks?
6. Robustness to Perturbations: How does the system handle perturbations during the trajectory execution?

**Robotics Focus:**

4

**Summary Of Paper:**

The authors present Dreamitate, a visuomotor policy learning framework that fine-tunes a video diffusion model on human demonstrations for dexterous manipulation tasks. The method generates videos of task execution conditioned on novel scenes and uses these videos to directly control robots. The key insight is using common tools to bridge the embodiment gap between human and robot manipulators. Dreamitate demonstrates superior generalization compared to behavior cloning methods, validated across four real-world tasks: rotation, scooping, sweeping, and shape pushing.

**Summary Of Recommendation:**

While the paper presents a novel approach to visuomotor policy learning using video generation and shows promising results, there are several areas that could benefit from further development. The focus on non-prehensile tasks and limited real-world complexity are significant weaknesses. Addressing these issues and expanding the evaluation to include prehensile and more realistic tasks would significantly strengthen the contribution.

---

### Official Review · Reviewer_w1Yn · 2024-07-21
**Review of submission 46**

**Originality:** 4
**Technical Quality:** 5
**Clarity Of Presentation:** 5
**Potential Impact:** 3
**Recommendation:** 3
**Confidence:** 5

**Review:**

# Clarity

The paper is well-written and easy to follow. The authors provide a strong introduction to the problem statement and convincingly explain how their method can improve performance over the current state-of-the-art.

# Strengths

The paper addresses a compelling problem: how robots can increase their training data without explicitly collecting expensive human demonstrations. The authors tackle this by employing video generation models that can create new videos in unseen circumstances using only a single initial state picture. They effectively combine the efficiency of behavior cloning in learning from data with video generation models to synthetically generate data, demonstrating a novel and promising approach.

# Weaknesses

My primary concern is that the paper relies heavily on having known tools to leverage generated videos. This constraint could significantly impact the method's ability to scale in the wild. Despite this limitation, the method works well within the parameters laid out by the authors.

The authors should also acknowledge the potential ambiguity in the initial image in their limitations section. For example, if there are two bowls with sprinkles and the objective is to scoop from only one, the method might struggle to determine the correct bowl. Additionally, the method's applicability to force-reliant tasks, such as cutting, is unclear and should be addressed as a potential limitation.

**Quality Of The Limitations Section:**

2

**Questions For Rebuttal:**

I think the authors should expand their limitations section to include that their method may not work effectively when there are multiple objects that can be interacted with similarly using the chosen tool. Additionally, their method may also fail with tasks that require precise force application, such as cutting or other force-dependent manipulations. Addressing these limitations would provide a more comprehensive understanding of the method's constraints and potential areas for improvement.

**Robotics Focus:**

4

**Summary Of Paper:**

This paper introduces a framework for visuomotor policy learning by fine-tuning a video generative model on human demonstrations. This approach leverages large-scale datasets of internet videos to generalize robot manipulation tasks across diverse environments. The framework synthesizes execution videos of tasks based on initial scene images, allowing robots to track and execute these tasks directly. Evaluated on four tasks, the method demonstrates better generalization compared to traditional behavior cloning approaches. The key insight is that using common tools bridges the gap between human and robot manipulation, enabling effective learning from minimal demonstrations.

**Summary Of Recommendation:**

The paper is well-written and introduces a compelling approach to enhance robot training data through video generation models, which can synthesize new videos from a single initial state image. This method effectively combines behavior cloning's efficiency with video generation to create synthetic data, demonstrating improvement over current state-of-the-art techniques. However, the reliance on known tools could limit scalability in real-world scenarios.

---

### Official Review · Reviewer_gdSH · 2024-07-26

**Originality:** 4
**Technical Quality:** 4
**Clarity Of Presentation:** 5
**Potential Impact:** 4
**Recommendation:** 3
**Confidence:** 4

**Review:**

The paper offers a compelling case for leveraging internet-scale video data for robust policy learning, though it also raises questions about scalability and practical implementation constraints.

**Strengths:**

- **Effective use of Internet-Scale Data:** The paper effectively demonstrates how leveraging pre-trained video generative models trained on internet-scale data can lead to more robust and generalizable policies.

- **Scalability:** By using human demonstrations instead of teleoperation data, the method significantly reduces the cost and complexity of data collection.

- **Interpretability:** The intermediate representation in the form of generated videos provides an interpretable output, which can be useful for understanding the robot's planned actions.

- **Agent-Agnostic:** The method's use of tool tracking makes the training pipeline agent-agnostic, allowing for transfer between different environments.

- **Performance:** The method demonstrates strong performance across multiple tasks, consistently outperforming the Diffusion Policy baseline, especially in generalizing to unseen objects and scenarios.

**Weaknesses:**

- **Scalability Concerns:** The lead over baselines appears to decrease as demonstration data increases (Fig. 8).

- **Requirement for Structured Environment:** The method requires calibrated camera extrinsics during fine-tuning and evaluation, which could be a practical limitation in unstructured environments. The method also requires CAD models of each tool, which may limit real-world applicability.

- **Single Task per Tool Assumption:** The approach seems to be designed for single-task policies per tool, which feels somewhat restrictive and may limit its flexibility in multi-task scenarios.

- **Baseline Comparison:** The baseline diffusion policy might not be optimally configured, especially regarding handling distractors, which raises concerns about the fairness of the comparison.

While the approach shows promise, I have some reservations about the method's long-term viability and scalability. The reliance on specific tools and the apparent diminishing returns with increased data raise questions about how far this method can be pushed. Furthermore, the computational demands of video generation models may be prohibitive for real-time applications. Despite these concerns, I believe this work represents an important step forward in leveraging large-scale generative models for robotics. It opens up new avenues for research and challenges us to think creatively about how we can harness the power of internet-scale data for robotic learning.

**Quality Of The Limitations Section:**

3

**Questions For Rebuttal:**

1. Why did you choose to track the tool instead of the end-effector? Why not, for example, use something like the UMI gripper (https://arxiv.org/abs/2402.10329) for data collection?

2. Why do some tasks have significantly more demonstrations than others? Did some tasks require more demonstrations for the video generation fine-tuning to work?

3. Are all policies single-task? If not, how is the task specified for the model?

4. What are the success criteria for the shape pushing task?

5. The results in Figure 8 show decreased performance with 2/3 of the data. What could be the reason for this decline? Was there significant variance in policy performance?

6. Are actions generated by simply transforming the tool pose from the camera frame to the robot frame? Is there any additional control mechanism, such as PID controllers or filters? Is the policy closed loop?

7. Your method segments out the background and hands to obtain object poses using Megapose. However, the diffusion baseline does not seem to have a corresponding mechanism to encourage generalization to various backgrounds. Could this difference have impacted the fairness of the comparison?

8. What computational resources are needed for training and evaluation? How long does the video generation take?

**Robotics Focus:**

4

**Summary Of Paper:**

This paper presents Dreamitate, a novel approach to visuomotor policy learning that leverages large-scale video generation models. The key idea is to fine-tune a video diffusion model on human demonstrations with a tool, track the tool's trajectory in the generated video, and translate the tool trajectory to robot actions. The authors demonstrate their approach on four manipulation tasks, showing improved generalization compared to a state-of-the-art baseline.

**Summary Of Recommendation:**

This paper presents an approach that shows promise in leveraging internet-scale data for robot learning. While I have some concerns about its scalability and real-world applicability, the results are impressive and the ideas are exciting, and I believe it could inspire important follow-up work. I think the robot learning community would benefit from engaging with these ideas, and I will consider revising my recommendation if my questions and concerns are addressed.

---

### Official Review · Reviewer_RxCE · 2024-07-29
**Interesting Paper and Idea but Baselines and Ablations are Missing**

**Originality:** 3
**Technical Quality:** 2
**Clarity Of Presentation:** 3
**Potential Impact:** 3
**Recommendation:** 2
**Confidence:** 3

**Review:**

Strengths:
1. The paper is well written and easy to understand
2, It addresses an important problem, that of utilizing free form human demonstrations obtained from an internet scale dataset to improve robot learning.
3. The method is straight forward and makes realistic assumptions

Weaknesses
1. **Lack of Baselines**: My biggest issue with the paper in its current state is the lack of baselines utilized by the authors. Works like Video as a Language for  https://arxiv.org/pdf/2402.17139, Video Language Planning https://arxiv.org/pdf/2310.10625 , UniPy https://proceedings.neurips.cc/paper_files/paper/2023/file/1d5b9233ad716a43be5c0d3023cb82d0-Paper-Conference.pdf have studied this problem before with varying degrees of success. The authors should utilize these baselines and compare how their method performs.
2. **Lack of Robot Foundation Models:** Currently, there are robot foundation model results in the paper. These are important as they serve as a skyline for the method, providing a metric for the potential performance if only robot data was used to train the behavior model.
3. **Lack of ablations:** The paper currently does not include any ablations and it is unclear which design decisions affect performance the most. This is necessary for completeness.

**Quality Of The Limitations Section:**

3

**Questions For Rebuttal:**

1. How does changing the camera angle affect the video model? Is it still robust to such differences?
2. What is the speed of policy execution? What is the inference speed of the video model?
3. Can you please describe the error modes of the work?

**Robotics Focus:**

4

**Summary Of Paper:**

The authors demonstrate how to use video prediction models as a way to train robot policies. First, an off-the-shelf video model is finetuned on a few human demonstrations of a tabletop task. Stereo cameras are used to collect demonstrations and subsequently, the generated pairs of videos are used to estimate the 3D position of a tool handled by the human. This 3D position subsequently serve as robot action which are executed. The method is tested on table top tasks like shape pushing, sweeping and scooping tasks.

**Summary Of Recommendation:**

I think the paper presents excellent ideas. However, I believe the experiments section needs improvement. Baselines and ablations are essential and are currently missing, which is why I am recommending a weak reject. With these improvements, I think this paper will be useful to many in the community.

---

### Author Rebuttal · Authors · 2024-08-13

The rebuttal files include the following:
1. A revised manuscript featuring additional baseline and ablation experiments, along with an expanded discussion on limitations.
2. An updated appendix containing further baseline and ablation experiments.
3. A collection of experiment videos for the set of four tasks.
4. Examples of video generations and robot executions from the no-pretraining ablation experiment.

---

### Decision · Program_Chairs · 2024-09-04

**Decision:**

Accept

**Comment:**

Strengths:
+ Dreamitate is a novel approach for using video-diffusion models in manipulation.
+ The approach is scalable, interpretable, and agent-agnostic.
+ The approach outperforms Diffusion Policies on real-world tasks.
+ The paper is well written.

Weaknesses:
- Missing video-conditioned baselines such as UniPi.
- Evaluations are missing prehensile tasks like grasping.
- Some details on the evaluation setup are missing.
- The method uses custom tools during data collection, which might hinder general applicability.
- The limitations section could benefit from a discussion on interactions with multiple objects, robustness to perturbations, and failure cases.

Post rebuttal:
Overall, the reviews are mixed; some key concerns regarding the generalizability of the approach remain. However, "video diffusion for robotics" is a relatively new field, so limitations could be addressed in future works.